# A Longitudinal Analysis of Alcohol Use Behavior among Korean Adults and Related Factors: A Latent Class Growth Model

**DOI:** 10.3390/ijerph18168797

**Published:** 2021-08-20

**Authors:** Suyon Baek, Eun-Hi Choi

**Affiliations:** 1Department of Nursing, College of Nursing and Health, Kongju National University, Gongju-si 32588, Korea; whitesy@kongju.ac.kr; 2Department of Nursing, College of Nursing, Eulji University, Uijeongbu-si 11759, Korea

**Keywords:** alcohol use behavior, Korean adults, latent class growth analysis, growth mixture modeling, Korean Welfare Panel Study

## Abstract

This study classified the changes in alcohol use behavior among Korean adults and explored the related factors. The study used data from the 4th (2009) to 14th (2019) waves of the Korean Welfare Panel Study. The subjects were 8267 adults aged 19–60 years. Latent class growth analysis was used to classify the latent classes of alcohol use behavior among Korean adults, and logistic regression analysis was performed to identify the specific factors that form the classes. Additionally, the 11-year trajectories of major variables associated with alcohol use behavior for the derived classes were analyzed using growth mixture modeling. Four classes were identified according to the trajectories of alcohol use behavior. There were statistically significant differences in the trajectories of depression, self-esteem, satisfaction in family relationships, and satisfaction in leisure activities according to the class of alcohol use behavior. In particular, self-esteem and satisfaction in family relationships indicated distinctly decreasing trajectories in the low- to moderate-risk class, which suggested the need for longitudinal analysis of the factors that influence alcohol use behavior. Moreover, it is recommended that interventions for the prevention of high-risk drinking target not only individuals but also family units.

## 1. Introduction

The harm caused by drinking alcohol is a major global health issue that negatively affects the health of individuals and has a negative socioeconomic impact [1]. According to the global status report on alcohol and health from the World Health Organization (WHO), everyone aged 15 years or older drinks 6.2 L of pure alcohol per year on average, which translates into 13.5 g of pure alcohol per day [2]. In Korea, the percentage of people who drink alcohol at least once a month is increasing, from 72.6% in 2005 to 75.3% in 2016 among men and from 36.9% in 2005 to 48.9% in 2016 among women [3]. In particular, the traditionally tolerant social atmosphere of drinking in Korea has recorded a continued increase in the percentage of high-risk drinkers [2]. The percentage of Korean drinkers who engage in heavy or binge drinking (≥7 servings for men and ≥5 servings for women) at least once a week is 47.7% [4], which is significantly higher than the worldwide average of 39.5% [1]. This problematic drinking has been reported to be the second biggest factor that negatively affects human health, shortening the lifespan of Koreans by 11.1 months [5], while it also causes various diseases, which in turn increase the financial burden of the health insurance system and socioeconomic loss.

Previous studies report that demographic factors such as age, marital status, educational level, and income level are related to alcohol use behavior [6]. In addition, intrapersonal factors such as self-esteem, suicidal ideation, and depression [7,8] or familial factors, including family relationships and family conflicts [9,10], are related to alcohol use behavior. Furthermore, it has been reported that social support [11] and leisure activities [12] influence alcohol use behavior. Thus far, studies in Korea on drinking have mostly examined adolescents, including middle school students [13,14,15] and college students in early adulthood [16,17,18]. Moreover, studies on adults have focused on the differences in alcohol use behavior between men and women [19,20] or identified the status using cross-sectional data. In other words, these cross-sectional studies were conducted to identify the influencing factors under the assumption that certain factors would act as the cause of alcohol use behavior, without considering the exact temporal relationship. However, some adults can control their alcohol use behavior without any problem or indicate a decreasing trajectory over time, whereas others exhibit increasing problems with their alcohol use behavior and continue to exhibit high levels of alcohol consumption. Accordingly, there is a need for studies to use longitudinal data to examine the trajectories of alcohol use behavior throughout the life cycle.

Despite this need, there are very few longitudinal studies on alcohol use behavior among Korean adults [21,22,23,24,25,26,27]. Moreover, most studies analyzed the longitudinal changes in alcohol use behavior as the dependent variable, but they used only data measured at one specific time point to analyze the influence of various factors on trajectories of alcohol use behavior. However, such studies may produce different results depending on the data measured at which time it is used. Even if the analysis is performed using data from the first point considering the temporal order of the cause and outcome, the effect on the dependent variable may vary according to subsequent changes in the subjects. Consequently, there are limitations when interpreting the findings of such studies. Accordingly, a study by Song et al. [28] suggested investigating the effects of changes in predictors on dependent variables.

The objective of this study was to use data from the Korean Welfare Panel Study (KOWEPS) to investigate the longitudinal trajectories of alcohol use behavior among Korean adults and examine the trajectories of factors identified as influencing factors through longitudinal analysis. The findings in this study could broaden the scope of understanding about the trajectories of alcohol use behavior in adults and serve as basic data for the development of alcohol use behavior improvement programs that are customized according to the pattern of alcohol use behavior among adults.

To achieve the objective as described above, the following research questions were established: (1) What is the number of latent classes according to the trajectories of alcohol use behavior among Korean adults, and how does the trajectory unique to each class appear? (2) What are the general characteristics that influence changes in the trajectories of alcohol use behavior among Korean adults? (3) What are the trajectories in depression, self-esteem, satisfaction in family relationships, and satisfaction in leisure activities according to the trajectories of alcohol use behavior among Korean adults?

## 2. Materials and Methods

### 2.1. Study Design

This is a secondary data analysis study designed to classify alcohol use behavior among Korean adults through latent class growth analysis (LCGA) and to identify the influence of general characteristics on each class and the trajectories of major variables.

### 2.2. Data Source

This study used data from KOWEPS, which is a nationally representative longitudinal survey that was conducted from 2006 to 2020 by the Korea Institute for Health and Social Affairs and the Seoul National University Institute of Social Welfare. For the samples, 30,000 households that participated in the 2006 National Living Conditions Survey were sampled using 2-stage stratified cluster sampling. Subsequently, these households were divided into low-income and regular-income households, and 7000 households (3500 households from each group) were selected using stratified cluster systematic sampling. Ultimately, first-year data were constructed using data from 7072 households. Half (50%) of the sampled households were designated as the low-income group—those with ≤60% of median income (≤120% of the poverty line), and the other half (50%) were designated as the regular-income group, those with >60% of median income, and the low-income group was over-sampled. Following the objective of the study, this study used data from the 4th (2009) to 14th (2019) waves of KOWEPS, since the alcohol use disorder identification test (AUDIT) for enquiring about alcohol use behavior was included in the survey starting from the 4th wave.

### 2.3. Subjects

Among 27,127 individuals who participated in the 1st (2006)–14th (2019) waves of KOWEPS, this study initially selected data from 18,054 individuals who responded at least three times to AUDIT items in the 4th (2009)–14th (2019) waves of KOWEPS for classification of groups according to the trajectories of alcohol use behavior over time using the LCGA method [29]. Of these individuals, those aged <19 or ≥61 years at the time of their first response to AUDIT were excluded. The legal drinking age in Korea is 20 years old or older; this corresponds to 19 years old or older in this study. Consequently, 8267 individuals were included in the analysis. The numbers of subjects evaluated in each wave are presented in Appendix A. Since it has been reported that a sample size of at least 300–500 is appropriate for LCGA [30], the sample size in this study satisfied the minimum sample size requirement.

### 2.4. Ethical Consideration

For KOWEPS, surveyors were trained before data collection, and prior consent was obtained from each participant. To use KOWEPS data for this study, a data use consent form was submitted, and upon approval, data with the removal of personally identifiable information were obtained and used in the analysis. This study received a review exemption from the Institutional Review Board of K University (IRB No: KNU_IRB_2020-26).

### 2.5. Selection and Definition of Study Variables

#### 2.5.1. Alcohol Use Behavior

Alcohol use behavior was measured using the WHO AUDIT scale. The AUDIT scale has been indicated to have appropriate sensitivity and specificity. It is a self-reporting alcohol abuse assessment scale used in many studies. The AUDIT scale consists of 10 items in three domains: three items regarding hazardous alcohol use, three items regarding alcohol dependence, and four items regarding problematic alcohol use. Eight items are graded on a 5-point scale (0–4 points), and two items are graded on a 3-point scale (0, 2, or 4 points). The AUDIT score is the sum of 10 items, ranging from 0 to 40 points. WHO considers an AUDIT score of ≥8 points to indicate alcohol dependence with hazardous alcohol use. Individuals with a score of 8–15 points are assessed to require simple advice for reducing dangerous alcohol use, a score of 16–19 points is assessed to require counseling and constant monitoring, and a score of ≥20 points is assessed to require treatment beyond diagnostic assessment for alcohol dependence.

#### 2.5.2. General Characteristics

The general characteristics identified included age, gender, employment status, family type (intact family, single-person family due to being single/separated/divorced/widowed, and grandparent/single-parent family), education level (high school graduate or below and college graduate or above), and low-income household status (low-income household ≤60% of median income, regular-income household >60% of median income).

#### 2.5.3. Depression

Depression was measured using the Center for Epidemiologic Studies Depression Scale-11 (CESD-11). Each of the 11 items is graded on a 4-point Likert scale [0 = rarely or none of the time (less than once a week) to 3 = most or almost all the time (at least five times a week)] with higher scores indicating higher levels of depression.

#### 2.5.4. Self-Esteem

Self-esteem was measured using the Rosenberg Self-Esteem Scale (RSES). Each of the 10 items is graded on a 4-point Likert scale (1 = Mostly not to 4 = Always so), with higher scores indicating higher self-esteem.

#### 2.5.5. Satisfaction in Family Relationships

Satisfaction in family relationships was measured using the question, “How satisfied are you with your family relationships?” (1 = very dissatisfied to 5 = very satisfied) This question was added from the 2nd wave onwards.

#### 2.5.6. Satisfaction in Leisure Activities

Satisfaction in leisure activities was measured using the question “How satisfied are you with your leisure activities?” (1 = very dissatisfied to 5 = very satisfied)

### 2.6. Data Analysis

Mplus 8.5 (Muthen & Muthen, Los Angeles, CA, USA) and SPSS 26.0 (IBM SPSS Statistics, New York, NY, USA) were used for data analysis in the study, in the following order according to the objective of the study.

First, the values of the variables were standardized. Rose mentioned that standardized scores should be used when comparing annually measured variables [31]. Accordingly, the scores for self-esteem, satisfaction in family relationships, and satisfaction in leisure activities were standardized (Z-score). For depression, raw scores were used because they were more familiar than standardized scores.

Second, LCGA was performed on alcohol use behavior repeatedly measured over time to identify the classes that indicated unique patterns [32]. To compare the models for determining the most suitable latent class, Akaike information criterion (AIC), Bayesian information criteria (BIC), sample size adjusted BIC (saBIC), Lo–Mendell likelihood ratio test (LMR), and bootstrap likelihood ratio test (BLRT) were used. With AIC and BIC, the lowest absolute values indicated good model fitness [32]. While AIC and BIC are influenced by the sample size, saBIC accounts for the sample size by adjusting for large sample size [33]. The LMR test compares the improvement in fit between models of k—1 and the k class and provides a *p*-value. That determines if there is a statistically significant improvement in fit that includes one more class. The BLRT uses bootstrap samples and provides a *p*-value that compares the increase in model fit between the k—1- and k class models [34].

Third, multinomial logistic regression analysis was performed with the input of general characteristics to identify the factors influencing the type of derived classes.

Fourth, growth mixture modeling was used to analyze the 11-year trajectories of depression, self-esteem, satisfaction in family relationships, and satisfaction in leisure activities, which are major variables associated with alcohol use behavior for the derived classes.

## 3. Results

### 3.1. Latent Class Model According to Trajectories of Alcohol Use Behavior

LCGA was performed to examine the latest classes according to the trajectories of alcohol use behavior (Table 1). AIC, BIC, and saBIC of the 4-Class model recorded low values, and LMR and BLRT indicated statistical significance. Therefore, the model was selected as the final model. In the 4-Class model, Classes 1, 2, 3, and 4 recorded classification rates of 7.31%, 5.93%, 13.25%, and 73.52%, respectively.

### 3.2. Intercept and Slope of Latent Classes

The intercepts and slopes of all latent classes were significant, except for the slope of Class 3 (Table 2). In Class 1, the mean AUDIT score decreased from 15.19 points at wave 1 to 6.10 points at wave 11. Accordingly, Class 1 was named the “moderate- to low-risk class.” In Class 2, the mean AUDIT score increased from 12.01 points at wave 1 to 12.01 points at wave 11. Accordingly, Class 2 was named the “low- to moderate-risk class.” In Class 3, the mean AUDIT score was 11.01 points at wave 1 and 10.27 points at wave 11. Accordingly, Class 3 was named the “stable moderate-risk class”. In Class 4, the mean AUDIT score was 2.77 points at wave 1 and 2.18 points at wave 11. Accordingly, Class 4 was named the “stable low-risk class.” The trajectories of different classes are presented as graphs in Figure 1.

### 3.3. General Characteristics Influencing Latent Classes

The multinomial logistic regression analysis results with the input of general characteristics according to latent classes are presented in Table 3. The regression model was significant (χ^2^ = 2231.105, df = 27, *p* < 0.001), while the Cox and Snell R^2^ values were 0.237 and that of Nagelkerke R^2^ was 0.289. Differences in general characteristics by latent class are presented in Appendix A.

When the “moderate- to low-risk class” (Class 1) was compared to the “stable low-risk class” (Class 4) as the reference class, the likelihood of belonging to Class 4 relative to Class 1 was significantly higher among subjects aged 19–29 years (OR = 0.141, *p* < 0.001) and 30–39 years (OR = 0.529, *p* < 0.001) than those aged 50–60 years. Additionally, the likelihood of belonging to Class 1 relative to Class 4 was significantly higher among males (OR = 11.447, *p* < 0.001) than females and among high school graduates or below (OR = 1.365, *p* = 0.001) than college graduates or above.

When the “low- to moderate-risk class” (Class 2) was compared with the “stable low-risk class” (Class 4) as the reference class, the likelihood of belonging to Class 2 relative to Class 4 was significantly higher among subjects aged 19–29 years (OR = 3.039, *p* < 0.001), 30–39 years (OR = 1.903, *p* < 0.001), and 40–49 years (OR = 1.448, *p* = 0.008) than those aged 50–60 years and among males (OR = 4.263, *p* < 0.001) than females. Meanwhile, the likelihood of belonging to Class 4 relative to Class 2 was significantly higher among unemployed subjects (OR = 0.550, *p* < 0.001) than among those who were employed. With respect to family type, the likelihood of belonging to Class 2 relative to Class 4 was significantly higher among single-person families (OR = 2.223, *p* < 0.001) and grandparent/single-parent families (OR = 2.952, *p* = 0.002) than intact families. Concerning education level, the likelihood of belonging to Class 2 relative to Class 4 was significantly higher among high school graduates or below (OR = 1.249, *p* = 0.036) than college graduates or above.

When the “stable moderate-risk class” (Class 3) was compared with the “stable low-risk class” (Class 4) as the reference class, the likelihood of belonging to Class 4 relative to Class 3 was significantly higher among subjects aged 19–29 years (OR = 0.255, *p* < 0.001) than those aged 50–60 years. Meanwhile, the likelihood of belonging to Class 3 relative to Class 4 was significantly higher among subjects aged 40–49 years (OR = 1.291, *p* = 0.003) than those aged 50–60 years. Moreover, the likelihood of belonging to Class 3 relative to Class 4 was significantly higher among males (OR = 14.525, *p* < 0.001) than females. Furthermore, the likelihood of belonging to Class 4 relative to Class 3 was significantly higher among unemployed subjects (OR = 0.483, *p* < 0.001) than those who were employed. Concerning the family type, the likelihood of belonging to Class 3 relative to Class 4 was significantly higher among single-person families (OR = 1.931, *p* < 0.001) than intact families and among high school graduates or below (OR = 1.436, *p* < 0.001) than college graduates or above. Regarding household income, the likelihood of belonging to Class 3 relative to Class 4 was significantly higher among low-income households (OR = 1.432, *p* = 0.004) than regular-income households.

### 3.4. Trajectories of Depression, Self-Esteem, Satisfaction in Family Relationships, and Satisfaction in Leisure Activities According to Latent Classes

The intercepts and slopes of the linear and quadratic terms of depression, self-esteem, satisfaction in family relationships, and satisfaction in leisure activities according to latent classes were analyzed (Table 4), and the trajectories were presented as graphs (Figure 2, Figure 3, Figure 4 and Figure 5).

The intercept and slopes of the linear and quadratic terms of depression were significant for all classes (*p* < 0.001). In the “moderate- to low-risk class”, depression was higher than in other classes, but it decreased over time. In the “low- to moderate-risk class”, depression was the highest among all classes. It decreased slightly over time and increased again to be higher than in the other classes at wave 11. In the “stable moderate-risk class”, depression was the lowest and lowest at wave 11 (Figure 2).

The intercepts and slopes of the linear and quadratic terms of self-esteem were significant for all classes (*p* < 0.001). In the “moderate- to low-risk class,” self-esteem was the lowest among all classes, but it increased over time. In the “low- to moderate-risk class,” self-esteem was higher than in other classes, but it decreased over time to be the lowest at wave 11 (Figure 3.).

The intercepts and slopes of the linear and quadratic terms of satisfaction in family relationships were significant for all classes (*p* < 0.001). In the “moderate- to low-risk class”, satisfaction in family relationships was the lowest, but it increased over time. In the “low- to moderate-risk class”, satisfaction in family relationships was the highest, but it decreased over time to be the lowest at wave 11 (Figure 4).

The intercepts and slopes of the linear and quadratic terms of satisfaction in leisure activities were significant for all classes (*p* < 0.001). In the “stable moderate-risk class”, satisfaction in leisure activities was the highest but decreased over time. In the “stable low-risk class”, satisfaction in leisure activities increased over time and indicated the highest score among all classes at wave 11 (Figure 5).

## 4. Discussion

This study examined the trajectories of alcohol use behavior among Korean adults between 2009 and 2019. The study also identified the general characteristics of the subjects by trajectory class and examined the trajectories of depression, self-esteem, satisfaction in family relationships, and satisfaction in leisure activities by class.

Four classes of alcohol use behavior were identified according to the trajectories of alcohol use behavior: “moderate- to low-risk class”, “low- to moderate-risk class”, “stable moderate-risk class”, and “stable low-risk class.” These findings were similar to “low-start, low-increase class”, “low-start, high-increase class”, and “high-start, stable class” identified in a study that investigated the trajectory classes of heavy drinking among African American adolescents [35]. Moreover, the findings were also similar to “stable high”, “early increasers”, “late increasers”, and “stable low” classes identified in a study that investigated the latent developmental trajectories of heavy drinking during the period between adolescence and early adulthood [36]. However, the “moderate- to low-risk class” was a unique class that was identified in this study, which could be attributed to the differences in the age of the subjects. In other words, the results recorded a decrease in high-risk alcohol use behavior with increasing age. A study on alcohol use among older adults aged ≥ 60 years also identified that the likelihood of belonging to a low-risk class, as compared to moderate- and high-risk classes, increased with age [37]. The findings in this study further indicated that individuals aged 19–49 years, as compared to those aged 50–60 years, had a higher likelihood of belonging to the “low- to moderate-risk class” or “stable moderate-risk class” than “ moderate- to low-risk class.” Therefore, it is necessary to plan and implement education for moderation in drinking and abstinence from drinking according to the age groups that belong to the high-risk group. Currently, the National Health Promotion Act in Korea stipulates that the national and local governments should provide education on the harm caused by excessive drinking, and by law, workplaces are required to provide education on moderation in drinking to their workers [38]. However, most regulations are mere recommendations and not mandatory. Therefore, it is essential to make such education mandatory and provide education for high-risk groups based on age.

In this study, males, as compared to females, were more likely to belong to the “moderate- to low-risk class”, “low- to moderate-risk class”, or “stable moderate-risk class” than the “stable low-risk class”. The “moderate- to low-risk class”, which had a decreasing trajectory in AUDIT scores, still recorded AUDIT scores that were higher than the “stable low-risk class”, which indicates that males have a higher tendency for high-risk alcohol use than females. Such findings also appeared in most studies on alcohol use behavior conducted in Korea [18,31]. Moreover, in a report on high-risk drinking rate (percentage of individuals among the general population aged ≥ 19 years who drank an average of at least seven servings per day for at least twice a week) between 2005 and 2017 [39], the high-risk drinking rate among Korean males remained steady at 21% (for females, the high-risk drinking rate—an average of at least five servings per day for at least twice a week—was 3.4% in 2005 and 7.2% in 2017). Such results can be attributed to the continuation of the drinking culture in Korean society, which tends to be more tolerant of alcohol use by males, and to the Korean social structure, where males, who are relatively more socially active than females, have more opportunities to be exposed to drinking environments.

Such results attributable to the social structure of Korea also appeared in the longitudinal changes in alcohol use behavior according to employment status. In other words, unemployed individuals were more likely to belong to the “stable low-risk class” than those who are employed. In a survey that investigated Koreans aged 19–59 years who drink at least once a month (monthly drinkers), major factors associated with drinking situations that lead to excessive drinking were identified as “when having company dinner or after-party with colleagues” and “entertaining clients” [40]. In other words, the workplace drinking culture in Korea that leads to *n* rounds of drinking with work colleagues is the cause of the increased likelihood of high-risk drinking by individuals participating in economic activities. However, in a survey on the harmful drinking experience rate among male monthly drinkers, “interference with work performance” accounted for 46.3% [40]; thus, attention should be paid to such workplace drinking culture at the corporate level. Accordingly, the time has come to create a workplace culture that excludes drinking and for companies to put greater effort into workplace drinking culture and drinking prevention education for workers.

Concerning the trajectories of alcohol use behavior according to family type, individuals belonging to a single-person family (unmarried, divorced, separated, or widowed) or grandparent/single-parent family were more likely to belong to the “low- to moderate-risk class” or “stable moderate-risk class” than the “stable low-risk class” as compared to those belonging to an intact family. Findings in a study suggested that problematic drinking is associated with health issues of single-person households because of the recent emergence of drinking culture targeting single-person households (drinking spirits alone, drinking beer alone, etc.) in Korea [41], while another study in Korea reported that drinking problems occur among single mothers because of their sense of responsibility as being the head of the household as a female while facing financial difficulties [42]. However, a cross-sectional study in Korea that used other nationally representative data [Korea National Health and Nutrition Examination Survey (KNHANES)] indicated that single-person households have a higher percentage of abstaining from drinking [43], while another cross-sectional study using KNHANES data reported that there were differences between men and women concerning the association between single-person households and heavy alcohol consumption [44]. Therefore, additional studies are required to determine how alcohol use behavior is influenced not only by family type but also by age and gender.

In this study, individuals with an education level of high school graduate or below were more likely to belong to the “low- to moderate-risk class”, “low- to moderate-risk class”, “moderate- to low-risk class”, or “stable moderate-risk class” than “stable low-risk class” and “stable moderate-risk class” than those with an education level of college graduate or above. Moreover, individuals belonging to low-income households were more likely to belong to the “stable moderate-risk class” than the “stable low-risk class” when compared to those belonging to regular-income households. These findings were consistent with a study reporting that lower education level increases the likelihood of belonging to the high-risk drinking group [37], as well as other studies reporting that a high-risk drinking rate [45], prevalence of alcohol use disorder [46], and alcohol-related health problems [47] are higher among low-income groups. Low socioeconomic status has been reported to influence problematic drinking because of a lack of alternatives and psychosocial causes. Low-income individuals face difficulties when participating in relatively diverse cultural activities, and consequently, they are more vulnerable to the temptations of inexpensive and easily accessible alcohol, which could lead to problematic drinking. In other words, individuals with lower socioeconomic status lack psychosocial resources for coping with stress, which is more likely to lead to problematic drinking [25]. Therefore, it is necessary to establish stress management programs, high-risk drinking prevention programs, and policies that are easily accessible by such groups. In 2000, the Korean government enacted the National Basic Living Security Act for the public assistance function as part of its social security system. The beneficiaries of this act are members of the low-income and poverty classes, who are the primary beneficiaries of community-based social welfare services. Private social welfare institutions and public community centers are responsible for accepting cases and providing management and services for difficulties faced by these individuals concerning their livelihood, housing, healthcare, and education. If intervention could also occur during such a process for screening and diagnosis of alcohol-related problems and stress management programs, then such high-risk drinking prevention policies could be more accessible to low-income groups.

Previous studies have examined the cross-sectional influence of major on the longitudinal trajectory of alcohol use behavior [25,35,36,37], whereas this study went a step further to analyze the longitudinal trajectories of depression, self-esteem, satisfaction in family relationships, and satisfaction in leisure activities according to the trajectory of alcohol use behavior among Korean adults. The results indicated that the intercepts and slopes of depression were statistically significant in all classes. The influence of depression on alcohol use behavior has consistently appeared in studies to date [15,38,46,48]. However, in this study, the intercept of depression was the lowest in the “stable moderate-risk class” and remained lower than all other classes over time. In a study that analyzed the association between alcohol consumption and depressive symptoms among elderly people in rural areas of Korea [49], a U-shaped curvilinear relationship was identified between alcohol use behavior (measured using AUDIT) and depressive symptoms (measured using BDI). In other words, BDI scores indicated a downward curve for AUDIT score of 0–14 points, which changed to an upward curve starting from an AUDIT score of 15 points (cutoff score to identify Korean problematic and high level of drinkers or alcohol abusers). Considering that the mean AUDIT score of the “stable moderate-risk class” was 10–11 points in this study, the results indicated a pattern similar to that of the study by Yun and Kim. Depression is a powerful influencing factor of alcohol use behavior, supported by intoxication and tension reduction theories [50]. However, some results that were inconsistent with those of previous studies suggest the need for additional in-depth studies on psychosocial factors other than depression to maintain high-risk alcohol use behavior among Korean adults.

The intercepts and slopes of self-esteem were statistically significant in all classes of alcohol use behavior. Self-esteem was the lowest in the “moderate- to low-risk class”, but it increased over time. Conversely, self-esteem was the highest in the “low- to moderate-risk class” but decreased over time to be the lowest at wave 11. Previous studies have not reported consistent results on the influence of self-esteem on alcohol use behavior. In other words, some studies reported that people with low self-esteem have a higher frequency and amount of alcohol consumption or have a higher likelihood of problematic drinking [15,51], whereas other studies reported that people with higher self-esteem are more likely to use alcohol [52] and that the influence of self-esteem could not be identified [53]. However, to date, studies have not investigated the longitudinal trajectory of self-esteem and tested the influence of self-esteem on alcohol use behavior at a specific time point. Conversely, this study analyzed the longitudinal trajectory of self-esteem, which has been investigated as a major factor in previous studies, according to the class of alcohol use behavior, and the results revealed that trajectories of self-esteem indicated statistically significant changes according to class. While such results do not confirm the influence of self-esteem on changes in alcohol use behavior, they clarify the association between the trajectory of self-esteem and alcohol use behavior.

The intercepts and slopes of satisfaction in family relationships were statistically significant in all classes of alcohol use behavior. Satisfaction in family relationships was the highest in the “low- to moderate-risk class”, but it decreased over time to be the lowest at wave 11. Conversely, it was the lowest in the “moderate- to low-risk class” but increased over time. Previous studies have also reported that alcohol use influences family cohesion and family conflict [16,17]. The drinking problem is not just a personal problem, but it could be referred to as a family disease that mutually affects family members who live together, while positive interactions between family members could be very important resources for preventing and reducing problematic drinking [25]. In England, the “Troubled Families” program was started in 2011, emphasizing that coping with addiction without considering the family is inappropriate (recited in [38]). This program included enhancement of parenting skills, support for family stability, and intensive case intervention to improve the lives of 120,000 families by 2015. This program recommends dealing with the problem as a family problem is more effective than separating and providing service to a single individual who has a drinking problem, and not doing so could cause families with complex problems to not receive effective assistance. However, policy strategies in Korea for improving drinking problems [54] focus mostly on the expanded establishment of alcohol counseling centers, occupational rehabilitation, and housing support, while policies focused on family support are still lacking. Therefore, Korea should also view families as a unit of intervention for drinking problems and focus more on the development of family-oriented interventions and policies, as well as support for individual family members.

Lastly, the intercepts and slopes of satisfaction in leisure activities were statistically significant in all classes of alcohol use behavior. Satisfaction in leisure activities was the highest in the “stable low-risk class” and lowest in the “low- to moderate-risk class” over time. Studies on the association between leisure activities and drinking among Korean adults could not be identified, while most studies outside of Korea on leisure activity and alcohol use reported that subjects who participate in sports activities tend to consume alcohol more often [12,55]. In a study that investigated the association between leisure activity and alcohol consumption among adolescents [56], an increase in structured leisure activities (engaging in artistic activities and cultural activities, etc.) significantly reduced alcohol consumption, whereas an increase in unstructured leisure activities (hanging out with friends, spending time in malls, etc.) significantly increased alcohol consumption. These results confirm that participation in high-quality leisure activities positively affects the reduction of alcohol consumption. In this study, satisfaction in leisure activities was measured with a single question, “How satisfied are you with your leisure activities?” Therefore, not knowing what type of leisure activity the subjects usually participated in and what their level of satisfaction was effected a limitation. Therefore, future studies on adult leisure activities that consider both quantitative and qualitative aspects are required. Based on such studies, a social atmosphere can be created and supported for high-quality adult leisure activities, which could be a method for preventing high-risk drinking among adults.

The limitations of this study are as follows. First, this study did not analyze the latent classes of alcohol use behavior according to gender. However, since the previous study was able to confirm the difference in AUDIT scores according to gender, we propose a future study to analyze the latent classes of alcohol use behavior according to gender and to identify the differences in influence factors. Second, we selected variables through a literature review; however, we were limited in not including enough variables. Therefore, in future research, it is necessary to increase the rationality in variable selection by basing it on the theoretical framework that explains alcohol use behavior.

## 5. Conclusions

This study used KOWEPS data to classify the longitudinal trajectories of alcohol use behavior among Korean adults and longitudinally analyzed the trajectories of major variables (depression, self-esteem, satisfaction in family relationships, and satisfaction in leisure activities) by class. The results indicated that there were four classes of trajectories of alcohol use behavior among Korean adults: “moderate- to low-risk class”, “low- to moderate-risk class”, “stable moderate-risk class”, and “stable low-risk class”. Subjects aged 19–49 years, males, employed, belonging to a single-person or grandparent/single-parent family, and have an education level of high school graduate or below were more likely to belong to the “stable moderate-risk class” or “low- to moderate-risk class.” Moreover, self-esteem and satisfaction in family relationships indicated distinctly decreasing trajectories in the “low- to moderate-risk class,” suggesting the need for longitudinal analysis of the factors that influence alcohol use behavior. Therefore, future longitudinal studies should include longitudinal analyses of dependent variables as well as various predictors. Furthermore, concerning alcohol use behavior among Korean adults, the findings in this study emphasize the need for mandatory drinking prevention education, intervention, and improvement of workplace drinking culture for men aged 20–40 years, who have been identified as a high-risk group, while also suggesting intervention for families, as well as individual drinkers.

## Figures and Tables

**Figure 1 ijerph-18-08797-f001:**
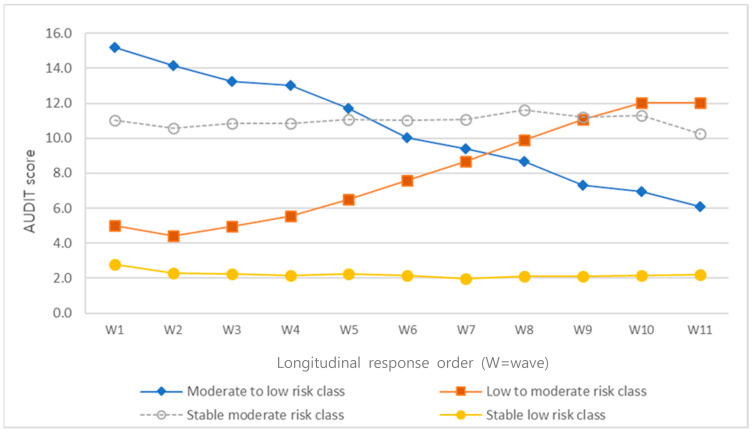
Latent Classes of Alcohol Use Behavior Trajectories.

**Figure 2 ijerph-18-08797-f002:**
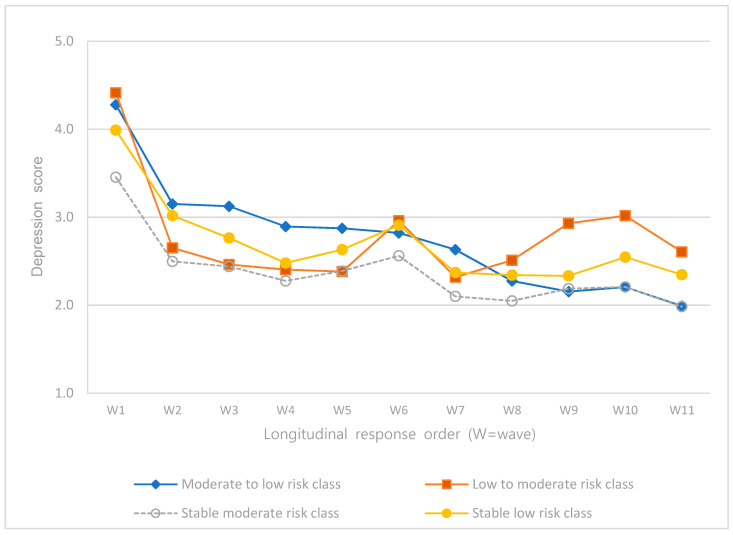
Trajectories of depression.

**Figure 3 ijerph-18-08797-f003:**
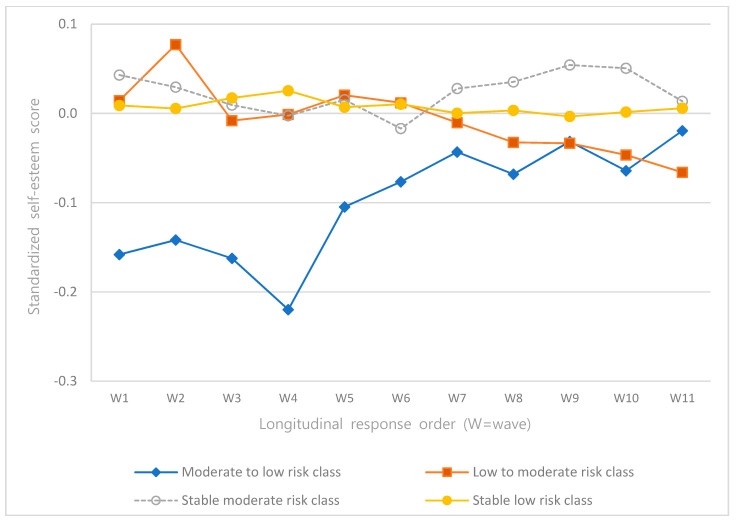
Trajectories of self-esteem.

**Figure 4 ijerph-18-08797-f004:**
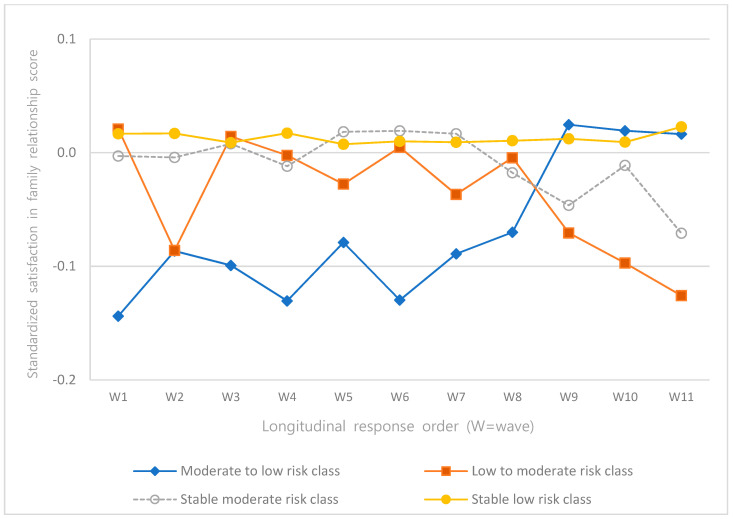
Trajectories of satisfaction in family relationships.

**Figure 5 ijerph-18-08797-f005:**
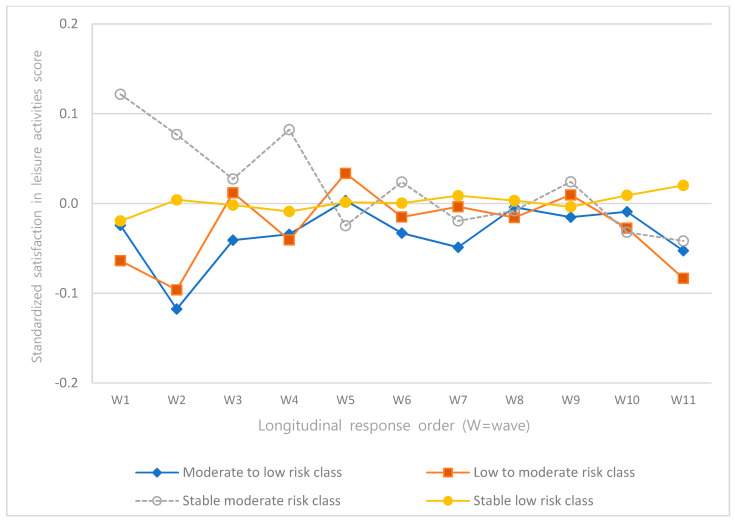
Trajectories of satisfaction in leisure activities.

**Table 1 ijerph-18-08797-t001:** Latent Class Model Fit for Trajectories of Alcohol Use Behavior (*n* = 8267).

Number of Classes	AIC	BIC	saBIC	LMR	BLRT	Estimated Probability for Trajectory Group(%)				
						1	2	3	4	5
1	381,294.250	381,406.570	381,355.725	*n*/a	*n*/a-	100.0				
2	379,272.308	379,406.688	373,345.310	<0.001	<0.001	22.49	77.51			
3	378,263.362	378,417.803	378,347.891	<0.001	<0.001	13.24	13.79	72.97		
4	377,734.311	377,909.811	377,830.366	0.001	<0.001	7.31	5.93	13.25	73.52	
5	377,218.619	377,415.180	377,326.201	0.108	<0.001	5.75	8.71	4.06	19.12	62.36

Abbreviations: AIC—Akaike information criterion; BIC—Bayesian information criterion; saBIC—sample size adjusted BIC; LMR—Lo–Mendell likelihood ratio test; BLRT— bootstrap likelihood ratio test.

**Table 2 ijerph-18-08797-t002:** Intercept and Slope of Each Class (*n* = 8267).

Parameter Estimate	Moderate to Low-Risk Class	Low to Moderate-Risk Class	Stable Moderate-Risk Class	Stable Low-Risk Class
	(Class 1, 7.31%)	(Class 2, 5.93%)	(Class 3, 13.25%)	(Class 4, 73.52%)
(Wave 1) Mean ± S.D	15.19 ± 5.82	5.02 ± 4.43	11.01 ± 5.10	2.77 ± 3.43
(Wave 11) Mean ± S.D	6.10 ± 4.48	12.01 ± 5.64	10.27 ± 4.15	2.18 ± 2.89
Intercept	14.808 (<0.001)	3.411 (<0.001)	10.594 (<0.001)	2.352 (<0.001)
Linear term	−0.900 (<0.001)	0.869 (<0.001)	−0.002 (0.974)	−0.030 (<0.001)

**Table 3 ijerph-18-08797-t003:** Multinomial Logistic Regression Analysis (*n* = 8267).

Characteristics		Categories		Comparison Group (Ref = Class 4)								
				Class 1			Class 2			Class 3		
				OR	95% CI	*p*	OR	95% CI	*p*	OR	95% CI	*p*
Age (ref = 50–60)		19–29		0.141	0.086–0.231	<0.001	3.039	2.285–4.042	<0.001	0.255	0.178–0.365	<0.001
		30–39		0.529	0.408–0.686	<0.001	1.903	1.425–2.543	<0.001	0.981	0.802–1.200	0.852
		40–49		0.834	0.680–1.023	0.082	1.448	1.100–1.905	0.008	1.291	1.090–1.530	0.003
Gender (ref = female)		Male		11.447	8.954–14.635	<0.001	4.263	3.471–5.237	<0.001	14.525	11.801–17.878	<0.001
Occupation (ref = employed)		Unemployed		0.930	0.729–1.187	0.562	0.550	0.427–0.709	<0.001	0.483	0.384–0.606	<0.001
Type of family (ref = intact families)		Single-person families		1.229	0.859–1.761	0.260	2.223	1.644–3.006	<0.001	1.931	1.487–2.507	<0.001
		Grandparent/single-parent families		0.529	0.161–1.740	0.295	2.952	1.509–5.775	0.002	1.257	0.623–2.540	0.523
Level of education (ref = college graduates or above)		High school graduates or below		1.365	1.129–1.650	0.001	1.249	1.015–1.538	0.036	1.436	1.234–1.672	<0.001
Household income (ref = regular-income) *		Low-income		1.272	0.960–1.686	0.094	1.266	0.921–1.740	0.147	1.432	1.123–1.824	0.004
				−2 Log Likelihood = 1195.105, χ^2^ = 2231.485, df = 27, *p* ≤ 0.001								
				Cox and Snell R^2^ = 0.237, Nagelkerke R^2^ = 0.289								

Class 1—moderate- to low-risk class; Class 2—low- to moderate-risk class; Class 3—stable moderate-risk class; Class 4—stable low-risk class; OR—adjusted odds ratio; CI—confidence interval.

**Table 4 ijerph-18-08797-t004:** Trajectories of Depression, Self-esteem, Satisfaction in Family relationship and Satisfaction in Leisure activities according to each class (*n* = 8267).

Parameter Estimate	Moderate- to Low-Risk Class			Low- to Moderate-Risk Class			Stable Moderate-Risk Class			Stable Low-Risk Class		
	Intercept	Linear Term	Quadratic Term	Intercept	Linear Term	Quadratic Term	Intercept	Linear Term	Quadratic Term	Intercept	Linear Term	Quadratic Term
Depression	3.098 *	−0.422 *	0.030 *	2.688 *	−0.436 *	0.041 *	2.331 *	−0.353 *	0.030 *	2.841 *	−0.453 *	0.038 *
Self-esteem	1.316 *	−0.323 *	0.031 *	1.482 *	−0.327 *	0.030 *	1.493 *	−0.341 *	0.030 *	1.491 *	−0.337 *	0.032 *
Satisfaction in family relationships	−0.600 *	−0.239 *	0.034 *	−0.534 *	−0.218 *	0.031 *	−0.536 *	−0.216 *	0.031 *	−0.513 *	−0.224 *	0.032 *
Satisfaction in leisure activities	−0.538 *	−0.236 *	0.034 *	−0.519 *	−0.221 *	0.032 *	−0.410 *	−0.262 *	0.036 *	−0.489 *	−0.240 *	0.035 *

* *p* < 0.001.

## Data Availability

Korea Welfare Panel Study. https://www.koweps.re.kr:442/data/data/list.do, accessed on 7 July 2021.

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
