# Peer review of "A Longitudinal Analysis of Alcohol Use Behavior among Korean Adults and Related Factors: A Latent Class Growth Model"

_ijerph, 2021, doi:10.3390/ijerph18168797_

Round 1

Reviewer 1 Report

  1. The authors should mention that the legal drinking age in South Korea is 20* years old. (*19 if the year in which you turn 20 has already commenced.) Since in other countries, the legal drinking age may be different. 
  2. Why were the sample weights not use? For instance, the individual longitudinal weight (P_WG_L or P_WS_L) or any of the other weights.
  3. Mention that the questions concerning family relationships were added from the 2nd wave onwards. 
  4. Why is the Literature Review not at the beginning?
  5. There needs to be an explanation concerning why the authors choose the variables for their analysis. I do not know the process and rationality used to choose the study variables, specifically self-esteem, satisfaction in family relationships, and leisure activities.
  6. Why use these variables self-esteem, satisfaction in family relationships, and leisure activities, especially when the KoWePS have other variables such as health, career, housing environment, and social network satisfaction. All these variables come from the Household Members survey, question three (Q 3), section C (i.e., Satisfaction & Views on Current Living Conditions), so why you did not use "Considering the above categories, how satisfied
    are you with your life?" (i.e., option h, Q 3, section C).
  7. (Also from the Household Members survey.) Why not use Q 9, section E (i.e., Daily Habits, Family Relationship, and Mental Health), "The following questions are intended to define how you perceive yourself."
  8. (Line 411) "Previous studies have examined the cross-sectional influence of major variables including depression and self-esteem on the longitudinal trajectory of alcohol-use behavior..." Would you please add references to these previous studies?
  9. Is "not knowing what type of leisure activity the subjects usually participated in" the only limitation (or weakness) of the study? 
  10. I would like to see some policy implications or clinical implications of this study's results.
  11. Please correct line 523 (Change ". Korea Health Statistics" to "Korea Health Statistics").

Author Response

I would like to extend my deep appreciation to you for review. I thankfully found this round of editing a good chance of improving the paper.

Reviewer 2 Report

Thank you for the opportunity to contribute to the peer review process for the original study submission manuscript entitled “A Longitudinal Analysis of Alcohol-Use Behavior among Korean Adults and Related Factors: A Latent Class Growth Model”. The manuscript is interesting, well written and points out relevant issues. The most important issue is related to the multivariable analysis in Table 3. My comments/questions are described below:

Line 100 (Subjects): Some points need to be clearer. How the gaps between the waves were treated? For example, who answered the 4th, 5th and 6th waves may have different changes in alcohol behavior then those who answered the 4th, 9th and 14th as the time between them varies. A supplementary table describing how many subjects were evaluated in each wave would help.

Line 165: The authors state that “LMR and BLRT provide significance probabilities for the class models” and the hypothesis. In fact, these tests compare the model with K classes to a model with (K-1) classes, differently of what is described. Please, amend this sentence.

Table 3:

- The most relevant comparison is the one that uses Class 4 as reference, the others may be excluded. This will shorten the results, which is too long.

- Are those OR crude? A multivariable multinomial logistic regression would be much better. Changing this, the discussion has to be reviewed.

Figures 2-5:

- Name y-axis and x-axis (or a footnote that W means the wave)

- The longitudinal analysis is superficial, there is no comparison of the trends among the 4 classes. Are there any significant difference of, p. ex., the “scores” for depression among the classes?

- Consider including 95% CI

Line 283: remove the word ‘figure’

Author Response

(The authors gave the same response as above.)

Round 2

Reviewer 1 Report

Good job!

Author Response

Thank you very much for your review.

Reviewer 2 Report

Thank you for the opportunity to contribute to your manuscript. All my comments and suggestions were adequately answered, except the one described below:

Line 175 still need some review:

These tests compare the model with K classes (alternative hypothesis) to a model with K-1 classes (null hypothesis), and the alternative hypothesis was that the model latent class model would be accepted the model fit is significantly better for the k classes model. The alternative hypothesis was accepted when the significance probability was ≤ 0.05.

Author Response

I would like to extend my deep appreciation to you for review. I thankfully found this round of editing a good chance of improving the paper.

I revised the part you mentioned as follows. I have also prepared a reference basis.

" The LMR test compares the improvement in fit between models of with k - 1 and the k class and provides a p value. That determine if there is a statistically significant improvement in fit that includes one more class. The BLRT uses bootstrap samples and provides a p value that compare the increase in model fit between the k - 1- and k class models [34]."